# The Role of Velocity-Based Training (VBT) in Enhancing Athletic Performance in Trained Individuals: A Meta-Analysis of Controlled Trials

**DOI:** 10.3390/ijerph19159252

**Published:** 2022-07-28

**Authors:** Xing Zhang, Siyuan Feng, Rui Peng, Hansen Li

**Affiliations:** 1Institute of Sports Science, College of Physical Education, Southwest University, Chongqing 400715, China; starz-94@foxmail.com; 2Laboratory of Genetics, University of Wisconsin-Madison, Madison, WI 53706, USA; siyuan.feng@wisc.edu; 3Department of Chemistry, McGill University, Montreal, QC H3A 0B8, Canada; pengr_uestc@163.com

**Keywords:** VBT, auto-regulation, resistance training, strength training, jump, sprint, strength endurance

## Abstract

Velocity-based training (VBT) is a rising auto-regulation method that dynamically regulates training loads to promote resistance training. However, the role of VBT in improving various athletic performances is still unclear. Hence, the presented study aimed to examine the role of VBT in improving lower limbs’ maximum strength, strength endurance, jump, and sprint performance among trained individuals. A systematic literature search was performed to identify studies on VBT for lower limb strength training via databases, including PubMed, Web of Science, Embase, EBSCO, Cochrane, CNKI (in Chinese), and Wanfang Database (in Chinese). Controlled trials that deployed VBT only without extra training content were considered. Eventually, nine studies with a total of 253 trained males (at least one year of training experience) were included in the meta-analysis. The pooled results suggest that VBT may effectively enhance lower limbs’ maximum strength (SMD = 0.76; *p* < 0.001; *I*^2^ = 0%), strength endurance (SMD = 1.19; *p* < 0.001; *I*^2^ = 2%), countermovement jump (SMD = 0.53; *p* < 0.001; *I*^2^ = 0%), and sprint ability (SMD of sprint time = −0.40; *p* < 0.001; *I*^2^ = 0%). These findings indicate the positive role of VBT in serving athletic training. Future research is warranted to focus on the effect of velocity loss of VBT on athletic performance.

## 1. Introduction

Resistance training (RT) is critical for enhancing athletic performance, including speed [1,2], agility [3], explosive strength [4], and even motor skills [5]. Traditional RT prescriptions are usually designed based on individuals’ 1RM (One repetition maximum) before starting an RT session [6]. Over the past decades, such a procedure has been widely accepted for prescribing training loads in RT [7]. However, athletes’ training state or performance is constantly changing due to numerous varying factors, such as diurnal biological variation, training fatigue, nutrient intake, and sleep. The factors can lead to up to 36% fluctuation in 1RM [8,9]. As daily fluctuation is rarely considered in traditional RT planning [10], the pre-designed training loads can be inappropriate and therefore reduce training benefits and even cause degeneration or injuries [11]. Therefore, a series of regulable and flexible RT methods, known as the auto-regulation methods, were developed to address the disadvantages of traditional RT.

Velocity-based training (VBT), a most advocated auto-regulation method, is defined as: “methods that use velocity to inform or enhance training practice” [10]. VBT uses objective velocity indices captured by monitors to timely evaluate training performance and dynamically regulate training loads [12]. The mean propulsive velocity (MPV) and the velocity loss (VL) are two standard indices for regulating training load [10]. The MPV is employed to select training weight, while VL is used to determine training repetitions during a set [10]. Thus, athletes have personalized training content that matches their daily readiness [10]. Moreover, the flexibility of VBT may bring extra advantages over traditional RT. For example, emerging evidence suggests that VBT usually requires a lower training volume to enhance strength in trained individuals [13]. Another study suggests that VBT results in a lower rate of perceived exertion (RPE) through dynamic adjustment. These facts highlight the value of VBT.

Owing to the mentioned advantages, VBT may serve sportspeople, particularly athletes who play rugby [14], football [15], basketball [16], and baseball [17], as they usually participate in various training regimes and frequent competitions. Lower energy consumption and fatigue in resistance training may help them better complete their training arrangements and reduce the risk of overtraining and injury [18]. In this context, exploring the role of VBT in improving various athletic performances may help understand VBT and advance its application. In fact, some studies have suggested the role of VBT in enhancing athletic performance, including strength [19], countermovement jump (CMJ), [20], and sprint ability [21]. A recent narrative review suggests that VBT can reduce neuromuscular fatigue and provide quality training to induce neuromuscular adaptations [22]. However, some scholars reported null results concerning VBT, including non-significant changes in CMJ [23], sprint [24], power test [23], and even degenerated sprint ability [25]. Since controlled trials involving athletes are usually challenged to recruit enough participants, quantitative analysis with a larger sample size may be essential to address the controversy and support the effectiveness of VBT in athletes. Hence, the purpose of the presented systematic review and meta-analysis was to examine the effectiveness of VBT in enhancing various athletic performances, including lower limbs’ maximum strength, strength endurance, jump, and sprint performance.

## 2. Materials and Methods

The current systematic review and meta-analysis was conducted in accordance with the recommendations of the Preferred Reporting Items for Systematic Reviews and Meta-Analysis (PRISMA).

### 2.1. Systematic Literature Search

A systematic literature search was conducted, and the following Chinese and English electronic databases were searched from inception to 5 September 2021: PubMed, Web of Science Embase, EBSCO, Cochrane, CNKI (China), and Wanfang Database (China). The searching strategy adapted for each database is presented in Table 1 (the search keywords were replaced by Chinese when searching Chinese databases).

### 2.2. Selection Criteria

Since we aimed to serve athletic training, studies on sportspeople who had at least a year of RT experience were considered. To obtain as many samples as possible, the current study included all types of controlled trials. Based on previous studies and reviews, one-repetition maximum (1RM), the maximum number of repetitions (MNR), countermovement jump (CMJ) and sprint time are frequently measured indices [26,27], so these indices were selected as our target outcomes. To avoid interferences from other training, we only included studies that had limited subjects to engage in extra training content.

In the current meta-analysis, the criterion of including and excluding studies accorded with the PICOS principle as following:-P (population): sportspeople who have at least one year of RT experience;-I (intervention): using VBT as lower limb strength training without extra training arrangement;-C (comparison): all types of controlled trials;-O (outcomes): the lower limbs’ 1RM, MNR, CMJ, and sprint time were measured in the training events;-S (study design): any types of control, including self-controlled designs that evaluated the effects of VBT on maximum strength, MNR, CMJ, and sprint time.

### 2.3. Literature Screening and Data Extraction

Study screening and selection were performed by two authors independently (HSL and XZ). The discrepancies were resolved by discussion or judgments from a third author (SYF). Data were extracted from included articles, including title, publication year, author name, study design, participant profile, sample size, intervention, intervention measure, measurements, and outcomes.

### 2.4. Quality Assessment

The methodological quality of the included studies was assessed by the PEDro scale (Physiotherapy Evidence-Based Database). According to the previous study, the PEDro scale has been evaluated to have high reliability and validity [28,29]. Items 2 to 11 were used to calculate the PEDro score. The methodology criteria were scored as: “Yes” (one point), “No” (zero points), or “Do not know” (zero points). The PEDro score of each selected study provided an indicator of the methodological quality (9–10 = excellent; 6–8 good; 4–5 = fair; <4 = poor), Two authors independently assessed the methodological quality of the included studies. The discrepancies were resolved by discussion or judgments from a third author (SYF).

### 2.5. Statistical Analysis

The meta-analysis was conducted using Review Manager 5.3 (Nordic Cochrane Centre, The Cochrane Collaboration, Copenhagen, Denmark). The random-effects model was used for the data synthesis of outcomes concerning 1RM, MNR, CMJ, and sprint time. Given the differences in the included studies (e.g., subjects, devices, and study environment), the standardized mean difference (SMD) was used to report the overall synthesis [30]. The level of significance was set as *p* < 0.05 and 95% confidence intervals [31]. The magnitude of effect was categorized as large (SMD > 0.8), medium (SMD = 0.5–0.8), small (SMD = 0.2–0.5), and trivial (SMD < 0.2) [32]. Statistical heterogeneity was assessed using *I*^2^ statistics [33,34]. The magnitude of heterogeneity for results was classified as low (<25%), moderate (25–75%), and high (>75%). A sensitivity analysis was conducted using the leave-one-out method to identify the source of the heterogeneity and further check the stability of the results. Funnel plots from Review Manager were used to generally identify publication bias in the pooled result [35].

## 3. Results

### 3.1. Selection of Studies

A total of 3581 studies were identified in the search. Thereafter, 3555 studies were excluded on duplication, title, and abstract. Sixteen studies were excluded for inappropriate controls or missing training background. One relevant study was not included due to unavailable full-text [36]. Finally, nine studies with 253 males were included in the current meta-analysis (Figure 1).

### 3.2. Study Characteristics

The studies were published between 2017 and 2021 (Table 2). All subjects were trained males (training years > 1 year). Most studies (*n* = 7) deployed an 8-week VBT intervention, and another two studies performed a 6-week [21] and a 7-week [37] VBT intervention, respectively. All studies employed squat (*n* = 9) as lower limb strength training. Regarding training arrangements, only one study [21] trained three times a week, and the rest of all the studies trained twice a week. In addition, most studies (*n* = 8) used T-Force Dynamic System (Spain) for measuring the velocity of the bar, and only one study [21] used GymAware Power Tool (Australia) instead. Regarding outcomes, all studies measured 1RM, seven studies reported CMJ, seven studies reported sprint time, including times of 10 m and 20 m, and four studies reported MNR.

### 3.3. Quality of the Included Studies

Eight studies were assessed as good quality, and only one study as fair quality according to the PEDro scale (Table 3). All studies were short in blinding, including blinding for subjects, therapists, and assessors.

### 3.4. Meta-Analysis

Nine studies investigated the effect of VBT on lower limbs’ maximum strength (Figure 2a and Appendix A). A statistically significant improvement in 1RM was found (SMD = 0.76, 95%CI: 0.58 to 0.94, *p* < 0.001), and no heterogeneity was observed (*I*^2^ = 0%).

Seven studies investigated the effect of VBT on sprint ability (Figure 2b and Appendix A). A statistically significant reduction in sprint time was found (SMD = −0.40, 95%CI: −0.57 to −0.23, *p* < 0.001), and no heterogeneity was observed (*I*^2^ = 0%).

Seven studies investigated the effect of VBT on jump performance (Figure 2c and Appendix A). A statistically significant improvement in CMJ was found (SMD = 0.53, 95%CI: 0.33 to 0.73, *p* < 0.001), and no heterogeneity was observed (*I*^2^ = 0%).

Four studies investigated the effect of VBT on lower limb strength endurance (Figure 2d and Appendix A). A statistically significant improvement in MNR was found (SMD = 1.19, 95%CI: 0.93 to 1.45, *p* < 0.001), and a trivial heterogeneity was observed (*I*^2^ = 2%).

### 3.5. Publication Bias

The funnel plots of 1RM, CMJ, and sprint time were nearly symmetrically distributed, indicating low risks of publication bias (Figure 3). The funnel plot of MNR demonstrated that more studies were distributed over the right, indicating a potential risk of bias.

## 4. Discussion

The current study aimed to examine the effectiveness of VBT in enhancing athletic performance in trained individuals. We found that VBT effectively enhanced lower limbs’ maximum strength, strength endurance, vertical jump, and sprint performance, which addressed the controversies in our identified studies. These findings offered quantitative evidence to support a recent narrative review that suggests the role of VBT in facilitating neuromuscular adaptations [22]. Another relevant meta-analysis has proved that VBT and traditional RT may lead to similar positive effects on several athletic performances [43]. However, their subjects not only underwent VBT. Some studies simultaneously deployed VBT and rugby [25] or running training [44]. Therefore, the benefits from other training contents could not be ruled out. By contrast, we included studies where participants only performed VBT, so the benefits for athletic performances, particularly for vertical jump and sprint, are more credible. Generally, our findings support the previous studies and highlight the effectiveness of VBT in improving various athletic performances.

### 4.1. Maximum Strength

The pooled result of the current meta-analysis revealed a medium positive effect of VBT on 1RM (SMD = 0.76, *p* < 0.001). This result suggests the effectiveness of VBT in developing lower limbs’ maximum strength. This result is not surprising because maximizing strength is the primary goal of RT [10], and any type of RT can theoretically enhance maximum strength. However, it is noteworthy that our participants were trained individuals. Unlike amateurs or untrained individuals, who can easily benefit from a random training arrangement, trained individuals need more accurate and appropriate training arrangements to make progress. Inappropriate training loads may not improve their maximum strength and even lower their performance below the baseline. However, all the included studies reported statistically significant improvement in maximum strength, which implies the effectiveness of VBT in improving maximum strength in trained individuals.

### 4.2. Strength Endurance

We observed a positive effect of VBT on MNR, and the effect size was large (SMD = 1.19, *p* < 0.001). This result suggests that VBT is an effective method for enhancing lower limb strength endurance in trained individuals. In RT, selective muscle hypertrophy is crucial for a specific training period. Theoretically, the increased strength endurance can be partially explained by positive changes in slow-twitch fibers or myosin heavy chain I (MHC-I) [45]. Traditional RT can be used to elicit these changes. Campos, et al. [46] conducted an 8-week intervention using the traditional RT method and found muscular endurance development accompanied by slow-twitch fibers hypertrophied. Likewise, Pareja-Blanco, et al. [23] checked the effects of an 8-week VBT on muscle structural and functional adaptations via magnetic resonance imaging, vastus lateralis biopsies, and kinematic test. Pareja-Blanco, et al. [23] found that VBT increased the cross-sectional area of slow-twitch fibers which, in turn, resulted in better strength endurance performance. A recent review has explained the selective hypertrophy of skeletal muscle in VBT. The velocity loss of VBT is positively associated with MHC-I percentage and negatively associated with myosin heavy chain IIX (MHC-IIX) percentage [10]. In other words, the velocity threshold can regulate the adaptions obtained in VBT. A greater velocity loss is more helpful for enhancing strength endurance.

### 4.3. Jump and Sprint Performance

We observed a medium positive effect of VBT on CMJ (SMD = 0.53, *p* < 0.001). Meanwhile, we found a decreased sprint time (SMD = −0.40, *p* < 0.001), indicating a positive effect of VBT on sprint performance. These results suggest the role of VBT in developing lower body power. Empirically, RT is not only fundamental for weightlifters, powerlifters, and bodybuilders but is also essential for other types of athletes to advance athletic performance [47,48]. Many reviews have proved traditional RT’s effectiveness in improving athletes’ jump and sprint performance [49,50,51]. Likewise, our results suggest the role of VBT in enhancing vertical jump and sprint performance in trained individuals, which implies that the velocity-based method can be a supplementary or alternative method to the traditional 1RM percentage-based method in resistance training.

An interesting point to note is that, although the overall result was positive, nearly half of our included studies reported null [23,24] and even adverse effects [23,38] in sprint tests. We checked their study designs and training prescriptions and found that all studies with null results adapted relatively greater velocity loss (>20%). As mentioned above, the velocity loss of VBT is highly related to selective muscle hypertrophy. A greater velocity loss could reduce MHC-IIX percentage and then inhibit its normal function [10]. This reduction may further negatively impact related explosive athletic performance, such as sprint, because MHC-IIX contributes to generating the highest contractile speed [45].

## 5. Limitation and Future Direction

Several limitations must be considered when interpreting the results of the current study. First, only male samples were involved in the relevant interventions. Thus, the current study’s findings may not be generalized to females. Second, we only used 1RM, MNR, CMJ, and sprint time to assess the effectiveness of VBT for developing athletic performance, due to the number of relevant studies. Other indicators, such as peak power and peak velocity, should be considered when more relevant articles appear [52,53,54,55]. Third, most of our included studies did not clearly describe their subjects’ backgrounds. Several studies restricted the intake of drugs and supplements, but subjects’ diet was not demonstrated. Daily diet may affect athletes’ responses to resistance training, which should be highlighted and controlled in future controlled trials. Moreover, the type of sports that subjects participated in, and their resistance training levels, may theoretically affect the training benefits of VBT. Unfortunately, we could only ensure the included subjects had over a year of training experience, while their resistance training level was unclear. Some included studies only roughly described their subjects as active sports science students, without illustrating their sports specialties. These limitations warrant further research on specific athlete populations.

## 6. Conclusions

The current study aimed to assess the effectiveness of VBT for developing athletic performance. Our meta-analysis results suggest that VBT intervention of 7–8 weeks may lead to distinctive improvements in lower limbs’ maximum strength, strength endurance, jump, and sprint performance among trained males. Since all the identified studies documented distinct maximum strength improvements, we recommend using the general VBT setting drawn from the studies (intensity of 1RM 50–85%, velocity loss of 0–45%, set of 3–5) as a reference for practical strength training. Moreover, the velocity loss of VBT may affect training benefits; for example, a relatively greater velocity loss may increase endurance but decrease explosive performance. Coaches and athletes should select the correct velocity loss threshold according to the training goals. The dose-response relationship between velocity loss and neuromuscular adaptation is not very clear yet, so we call for more focus on the role of velocity loss on athletic performance.

## Figures and Tables

**Figure 1 ijerph-19-09252-f001:**
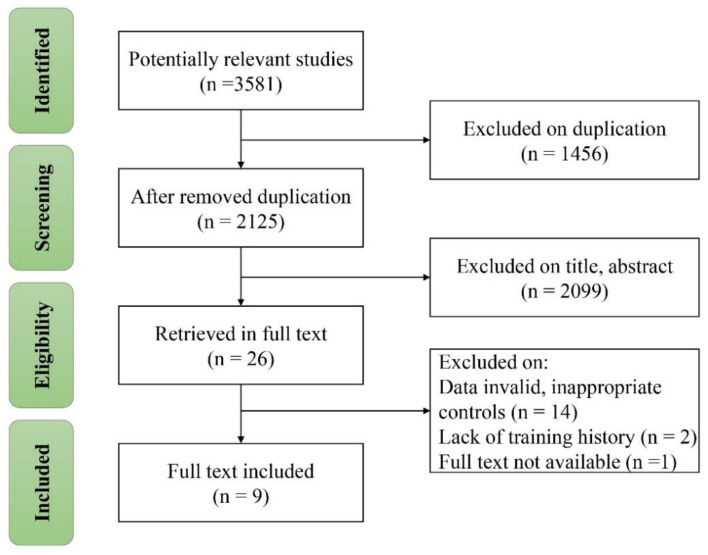
Flow diagram of screening and selection of studies.

**Figure 2 ijerph-19-09252-f002:**
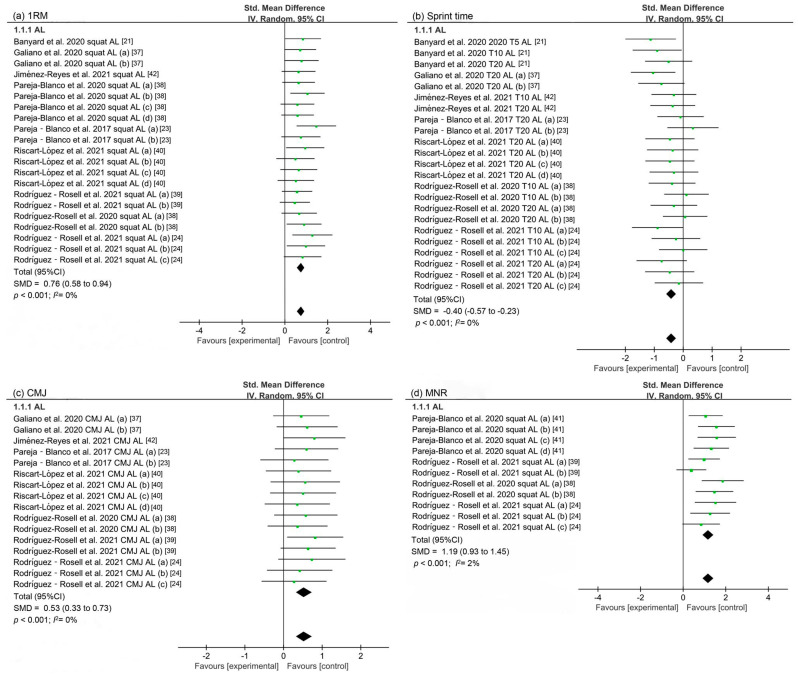
The effect of VBT on athletic performances. (**a**–**d**) indicate pooled results of 1RM, sprint time, countermovement jump (CMJ), and maximum number of repetitions (MNR), respectively [21,23,24,37,38,39,40,41,42].

**Figure 3 ijerph-19-09252-f003:**
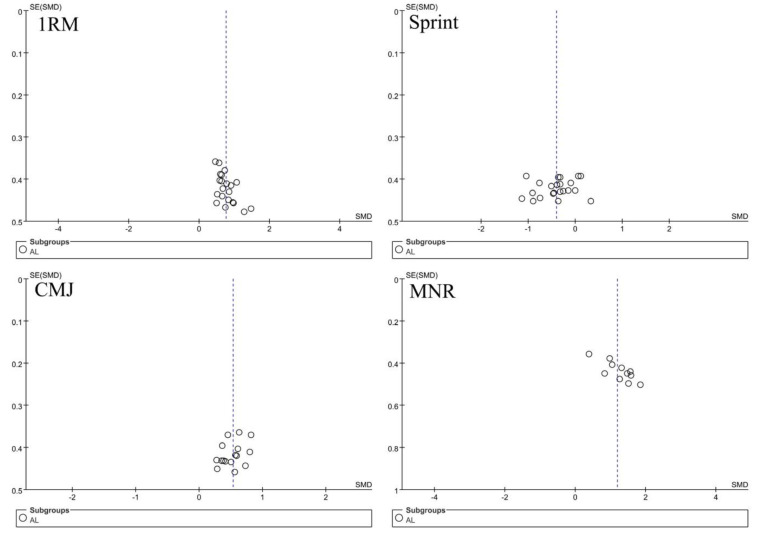
Funnel plots for the assessment of publication bias.

**Table 1 ijerph-19-09252-t001:** Searching strategy for the study inclusion.

Steps	Searching Command	Field
#1	Velocity-based training OR VBT OR velocity-based resistance training OR VBRT OR velocity loss OR VL	Title or abstract
#2	Strength OR one repetition maximum OR 1RM OR power OR countermovement jump OR CMJ OR vertical jump OR sprint OR speed test	Title or abstract
#3	#1 AND #2	

**Table 2 ijerph-19-09252-t002:** Characteristics of studies included in the present study.

Authors	Samples	The Level of Athletes	Measuring Tool	Intervention (Program)	Outcomes	Training Settings
Rodríguez-Rosell et al., 2021 [24]	Sample: 33; Gender: male; Age: 22.8 ± 3.9 years; Experience: 1~3 years; Supplements: unknown.	Unknown	T-Force System	8 weeks (LP)	1RM, CMJ, T10, T20, MNR	Event: squat; Intensity: 55~60% 1RM; Repetition: VL10 to 45%; Set: 3 sets; Inter-set recovery: 4 min; Frequency: twice a week; Session: 16 sessions.
Rodríguez-Rosell et al., 2020 [38]	Sample: 25; Gender: male; Age: 22.5 years; Experience: 1~3 years; Supplements: unknown.	Physically active sport science students	T-Force System	8 weeks (LP)	1RM, CMJ, T10, T20, MNR	Event: squat; Intensity: 70~85% 1RM; Repetition: VL10 to 30%; Set: 3 sets; Inter-set recovery: 3–5 min; Frequency: twice a week; Session: 16 sessions.
Rodríguez-Rosell et al., 2021 [39]	Sample: 32; Gender: male; Age: 23.5 years; Experience: 1~3 years. Supplements: no.	Physically active sport science students	T-Force System	8 weeks (LP and UP)	1RM, CMJ, MNR	Event: squat; Intensity: 50~80% 1RM; Repetition: VL20%; Set: 4 sets; Inter-set recovery: 4 min; Frequency: twice a week; Session: 16 sessions.
Riscart-López et al., 2021 [40]	Sample: 33; Gender: male; Age: 23.2 years; Experience: 1.5~4 years; Supplements: no.	Physically active sport science students	T-Force System	8 weeks (LP, UP, RP and CP)	1RM, CMJ, T20	Event: squat; Intensity: 50~85% 1RM; Repetition: VL20%; Set: 3 sets; Inter-set recovery: 4 min; Frequency: twice a week; Session: 16 sessions.
Pareja-Blanco et al., 2020 [41]	Sample: 55; Gender: male; Age: 24.1 ± 4.3 years; Experience: 1.5~4 years; Supplements: no.	Unknown	T-Force System	8 weeks (LP)	1RM, MNR	Event: squat; Intensity: 70~85% 1RM; Repetition: VL0 to 40%; Set: 3 sets; Inter-set recovery: 4 min; Frequency: twice a week; Session: 16 sessions.
Pareja-Blanco et al., 2017 [23]	Sample: 22; Gender: male; Age: 22.7 ± 1.9 years; Experience: 1.5~4 years; Supplements: no.	Physically active sport science students	T-Force system	8 weeks (LP)	1RM, CMJ, T20	Event: squat; Intensity: 69~75% 1RM; Repetition: VL20 to 40%; Set: 3 sets; Inter-set recovery: 4 min; Frequency: twice a week; Session: 16 sessions.
Jiménez-Reyes et al., 2021 [42]	Sample: 13; Gender: male; Age: 23.1 ± 4.1 years; Experience: >2 years; Supplements: no.	Physically active sport science students	T-Force System	8 weeks (LP)	1RM, CMJ, T10, T20	Event: squat; Intensity: 50~80% 1RM; Repetition: VL10 to 15%; Set: 3~4 sets; Inter-set recovery: 3~5 min; Frequency: twice a week; Session: 16 sessions.
Galiano et al., 2020 [37]	Sample: 28; Gender: male; Age: 22.9 years; Experience: >1.5 years; Supplements: no.	Unknown	T-Force System	7 weeks (CP)	1RM, CMJ, T20	Event: squat; Intensity: 59~85% 1RM; Repetition: VL5 to 20%; Set: 3 sets; Inter-set recovery: 3 min; Frequency: twice a week; Session: 14 sessions.
Banyard et al., 2020 [21]	Sample: 12; Gender: male; Age: 25.5 ± 5.0 years; Experience: >2 years; Supplements: unknown.	Unknown	GymAware Power Tool	6 weeks (UP)	1RM, T5, T10, T20	Event: squat; Intensity: 59~85% 1RM; Repetition: 5 repetitions; Set: 5 sets; Inter-set recovery: 2 min; Frequency: three times a week; Session: 18 sessions.

**Note:** T10, 10 m sprint time; T20, 20 m sprint time; 1RM, One Repetition Maximum; CMJ, Countermovement Jump; MNR, Maximal Number of Repetitions; VL, Velocity loss; LP, Linear programming; UP, Undulating programming; RP, Reverse programming; CP; Constant programming; Supplements, to take drugs, medications, or supplements.

**Table 3 ijerph-19-09252-t003:** Quality assessment of the included studies.

Studies	Pedro Item											Assessment
	1	2	3	4	5	6	7	8	9	10	11	
Rodríguez-Rosell et al., 2021 [24]	Yes	1	-	1	-	-	-	1	1	1	1	good
Rodríguez-Rosell et al., 2020 [38]	Yes	1	-	1	-	-	-	1	1	1	1	good
Rodríguez-Rosell et al., 2021 [39]	Yes	1	-	1	-	-	-	1	1	1	1	good
Riscart-López et al., 2021 [40]	Yes	1	-	1	-	-	-	1	1	1	1	good
Pareja-Blanco et al., 2020 [41]	Yes	1	-	1	-	-	-	1	1	1	1	good
Pareja-Blanco et al., 2017 [23]	Yes	1	-	1	-	-	-	1	1	1	1	good
Jiménez-Reyes et al., 2021 [42]	Yes	1	-	1	-	-	-	1	1	1	1	good
Galiano et al., 2020 [37]	Yes	1	-	1	-	-	-	1	1	1	1	good
Banyard et al., 2020 [21]	Yes	-	-	1	-	-	-	1	1	1	1	fair

**Note:** Items: 1. Eligibility criteria were specified. 2. Subjects were randomly allocated to groups (in a crossover study, subjects were randomly allocated an order in which treatments were received). 3. Allocation was concealed. 4. The groups were similar at baseline regarding the most important prognostic indicators. 5. There was blinding of all subjects. 6. There was blinding of all therapists who administered the therapy. 7. There was blinding of all assessors who measured at least one key outcome. 8. Measures of at least one key outcome were obtained from more than 85% of the subjects initially allocated to groups. 9. All subjects for whom outcome measures were available received the treatment or control condition as allocated, or, where this was not the case, data for at least one key outcome was analyzed by “intention to treat.” 10. The results of between-group statistical comparisons are reported for at least one key outcome. 11. The study provides both point measures and measures of variability for at least one key outcome.

## Data Availability

The data is available upon request from the corresponding author.

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
