# Peer review of "The Role of Velocity-Based Training (VBT) in Enhancing Athletic Performance in Trained Individuals: A Meta-Analysis of Controlled Trials"

_ijerph, 2022, doi:10.3390/ijerph19159252_

Round 1
Reviewer 1 Report
The authors should be commended for clarity and depth in describing the process (materials and methods) of the meta-analysis. Good inclusion of a visual flow chart (Figure 1) to help guide the reader.
It was noted that the discussion in lines 267-275 addressed the comment in line 51 which I was looking for.
The paper was written clearly with a narrow focus to effectively examine the research topic well.
Author Response
We thank the reviewer for assessing our work and giving us high marks.

Reviewer 2 Report
I would like to thank the authors for writing this study.
It addresses an important and very topical aspect of VBT or VBRT training and provides new knowledge about it.
It is good that meta-analyses are appearing that really provide new and solid evidence about this area of research.
As the only point to consider, I would ask the authors to make the figures more "attractive", this could increase the interest of the readers, since the results section is difficult to interpret with so many tables and figures that are scarce and not very "attractive". From my point of view, data visualization is fundamental to achieve an excellent work and, really, few authors take this into account when writing their studies.
Otherwise, good work.
Author Response
We thank the reviewer for assessing our work and the encouraging words. We have combined the figures to make them more concise.

Reviewer 3 Report
The work is interesting, but the methodical part should certainly be supplemented with information related to the diet, which will find a way to influence the analyzed data and the results. Even the authors themselves are aware of this in 235 line "Lower energy and time consumption in resistance training may reduce physical and mental fatigue and therefore benefit other subsequent training.
Line 82- intervention- What was the quality of the diet? What was the nutritional status?
Line 89, 177, 191, 204- without coma
Line 276- capital
Author Response
We thank the reviewer for his/her time into this work. Please check responses and revisions below.

Reviewer 4 Report
This reviewer would like to thanks to the editor the opportunity to review the paper entitled “The role of velocity-based training (VBT) in enhancing athletic performance in trained individuals: a meta-analysis of controlled trials”. The aim of this systematic review with meta-analisys was “to examine the effectiveness of VBT in enhancing various athletic performances, including lower limbs’ maximum strength, strength endurance, jump, and sprint performance”. Once I have read the manuscript carefully, I would like to express my congratulations to the authors, since they conducted a great research. Also, a great effort. The manuscript has a great accuracy, and it supposes a step forward in relation with strength training based on velocity. However, I have some doubts about some specific points of the manuscript that I really think that it should be clarified.
· Line 14: Add the Word “review”
· Line 58: You should clarify what kind of jump are you talking about.
· I really recommend to the author to include a description table with as many details as possible in relation with the training characteristics in each of the studies analysed. The table should include training program specifications, number of repetitions, time rest among sets, barbel velocity to perform the exercises, %1RM…It allows to the readers a better understanding of the results in your manuscript.
· The authors indicate that all the participants had at least one year of training experience. I consider that this information about the participants is not complete, consequently, I would like the author include more information related with they such as: a) What is the level of participants in strength training? b) It possible to know what is the sport that the participants usually practice (i.e., powerlifting, CrossFit, running‚). Also, what is the sport category of the athlete? c) In relation to previous questions, I would like to know if the sport modality practiced, also the level of athletes could have an influence on the results? Please, you should include this information to clarify all this points.
· I would like to know how the authors can explain the wide range of 1RM (between 50-85%) utilized in the papers analysed since the training program, also the results are totally different depend on the %RM in each training program. Please, try to clarify this information since it is important to know properly the intensity or workout in order to obtained improvements in the variables mentioned in your manuscript.
· You should include more information about the advantages that this method has in comparison with traditional training methods. Among others, a) you should include if this method has a lower injury risk, also if it induces less fatigue in the athletes; b) How this method can modify the training planification of the athlete (i.e., less time to require to complete the workout, a greater recovery between sessions…).
· I totally recommend including a practical application’s point where the authors explain carefully how the coach and the athlete can apply the results of this manuscript in daily training routines. Also, the benefit of this method can have if it is included in the routines of different sport modalities.
Author Response
We thank the reviewer for assessing the work and the encouraging words. Please check revisions in blue in the revised manuscript.
